# RNA-Seq Analysis Identifies Differentially Expressed Genes in the *Longissimus dorsi* of Wagyu and Chinese Red Steppe Cattle

**DOI:** 10.3390/ijms24010387

**Published:** 2022-12-26

**Authors:** Guanghui Li, Runjun Yang, Xin Lu, Yue Liu, Wei He, Yue Li, Haibin Yu, Lihong Qin, Yang Cao, Zhihui Zhao, Xibi Fang

**Affiliations:** 1College of Animal Science, Jilin University, Changchun 130062, China; 2College of Coastal Agricultural Sciences, Guangdong Ocean University, Zhanjiang 524000, China; 3Branch of Animal Husbandry, Jilin Academy of Agricultural Sciences, Changchun 130062, China

**Keywords:** beef cattle, RNA sequencing, differentially expressed genes, meat quality traits, fat metabolism

## Abstract

Meat quality has a close relationship with fat and connective tissue; therefore, screening and identifying functional genes related to lipid metabolism is essential for the production of high-grade beef. The transcriptomes of the *Longissimus dorsi* muscle in Wagyu and Chinese Red Steppe cattle, breeds with significant differences in meat quality and intramuscular fat deposition, were analyzed using RNA-seq to screen for candidate genes associated with beef quality traits. Gene Ontology (GO) and Kyoto Encyclopedia of Genes and Genomes (KEGG) enrichment analysis showed that the 388 differentially expressed genes (DEGs) were involved in biological processes such as short-chain fatty acid metabolism, regulation of fatty acid transport and the peroxisome proliferator-activated receptor (PPAR) signaling pathway. In addition, crystallin alpha B (*CRYAB*), ankyrin repeat domain 2 (*ANKRD2*), aldehyde dehydrogenase 9 family member A1 (*ALDH9A1*) and enoyl-CoA hydratase and 3-hydroxyacyl CoA dehydrogenase (*EHHADH*) were investigated for their effects on intracellular triglyceride and fatty acid content and their regulatory effects on genes in lipogenesis and fatty acid metabolism pathways. This study generated a dataset from transcriptome profiling of two cattle breeds, with differing capacities for fat-deposition in the muscle, and revealed molecular evidence that *CRYAB*, *ANKRD2*, *ALDH9A1* and *EHHADH* are related to fat metabolism in bovine fetal fibroblasts (BFFs). The results provide potential functional genes for maker-assisted selection and molecular breeding to improve meat quality traits in beef cattle.

## 1. Introduction

Beef is a popular meat, known for being rich in nutrients that are important for antioxidant and anti-inflammatory responses as well as nerve, muscle, retinal, immune and cardiovascular function [1]. In recent years, meat characteristics, particularly fat deposition, have become an important factor influencing consumers’ meat purchasing decisions [2]. Livestock lipid metabolism is mainly affected by heredity, feeding and the external environment [3]. With the development of modern molecular breeding techniques, such as cell engineering and molecular markers, modern beef cattle breeding combines conventional selective breeding with molecular biology, bioinformatics and computer information technology. Screening and verification of functional genes has become important for molecular breeding to improve meat quality traits. In addition, the cellular and genetic mechanisms of different fat deposition sites involve complex and highly coordinated gene expression programs. Therefore, lipid metabolism has always been a topic that offers both challenge and interest in livestock animal research.

Transcriptomics was the first (and is now the most widely used) molecular technology in basic research, clinical diagnosis and drug development [4]. In recent years, with the rapid development of RNA-seq, it has been applied in the livestock industry. RNA-seq analyses have been used to determine large numbers of candidate genes, new transcripts, single nucleotide polymorphisms (SNPs), and regulatory networks for different species and for tissues of different animals such as pigs, cattle and sheep [5]. Understanding the transcriptomes of livestock animals is critical for explaining the function of their genomes, revealing the molecular makeup of cells and tissues and exploring development and disease. With transcriptomics now applied broadly to livestock studies [6,7,8,9,10,11,12,13], several RNA-seq analyses have already been performed in the Wagyu and other breeds in the U.S. and European regions to select functional genes for improved meat quality [14,15]. However, only a few studies have been applied to screen for differences between functional genes in Wagyu and the yellow cattle of Asia.

Chinese Red Steppe cattle are a native Chinese breed used for meat and dairy, mainly distributed in the northeast of China [11]. They have unique features, such as disease resistance and better meat quality, compared to other local Chinese cattle. They are popular among meat consumers because of the unique flavor of their meat. However, Chinese Red Steppe cattle have a lower intramuscular fat (IMF) content than foreign commercial beef cattle, which limits the beef quality and economic benefits. The Wagyu is a Japanese beef cattle breed derived from native Asian cattle. The most distinctive feature of Wagyu beef is its beautiful marbling, due mainly to its high intramuscular fat content, which improves the overall taste [16]. A previous meat quality assessment found that Wagyu beef is high in fat, which is essential for improving the texture of the meat [16]. Studies have shown that genetic differences in meat quality are expressed as inter-varietal differences [17]. Therefore, the two breeds of cattle are perfect models for screening functional genes associated with fat deposition affecting meat quality traits. In addition, the study of these model breeds could reveal novel key genes and regulatory networks that are involved in regulating fat deposition in muscle tissue, leading to further improvement of meat quality traits in Asian cattle. 

In the present study, the *Longissimus dorsi* muscles of Wagyu and Chinese Red Steppe cattle were used to screen differentially expressed genes (DEGs) by RNA-seq, and the functions of candidate genes related to meat traits in lipid metabolism were analyzed. These results revealed novel functional genes for the molecular breeding of cattle, which will be useful for further study on the meat quality of different breeds.

## 2. Results

### 2.1. Analysis of DEGs in Longissimus dorsi Muscle between Cattle Breeds

The details of the transcriptome sequencing data are reported in our previous study on the alternative splicing comparative analysis of the *Longissimus dorsi* muscle in Wagyu and Chinese Red Steppe cattle [11]. The data were submitted to the Gene Expression Omnibus (GEO) of the National Center for Biotechnology Information (NCBI) [18] and are accessible through GEO Series accession number GSE161967 (https://www.ncbi.nlm.nih.gov/geo/query/acc.cgi?acc=GSE161967; accessed on 1 May 2022).

A total of 388 DEGs were obtained via transcriptome expression differential gene analysis (Log_2_ FC > 0.58 or < −0.585, false discovery rate (FDR) < 0.05). The enrichment heat map is shown in Figure 1A, within which the key genes are reportedly associated with meat quality traits including insulin-like growth factor 2 (*IGF-2*), acyl-CoA oxidase 1 (*ACOX1*), angiopoietin like 4 (*ANGPTL4*) and solute carrier family 27 member 5 (*SLC27A5*). A large number of novel candidate genes connected to bovine muscle development and lipid metabolism were also identified: agouti signaling protein (*ASIP*), CD44 molecule (*CD44*), syndecan-3 (*SDC3*), agrin (*AGRN*), fructose-bisphosphatase 1 (*FBP1*), glutamic-oxaloacetic transaminase1, *(ACTC1*), crystallin alpha B (*CRYAB*), and ankyrin repeat domain 2 (*ANKRD2*). Among the DEGs, 205 genes were up-regulated in the *Longissimus dorsi* of Wagyu compared with Chinese Red Steppe cattle, and 183 genes were down-regulated. The details of the top 10 up- and down-regulated DEGs are shown in Table 1 and Table 2, respectively. Notably, 22 genes were uncharacterized in the database, the details of which are in Appendix A.

### 2.2. DEG Participation in Biological Processes Related to Lipid Metabolism

To understand their putative functions, 388 DEGs in Wagyu were mapped to the Gene Ontology (GO) and Kyoto Encyclopedia of Genes and Genomes (KEGG) orthology (KO) databases for analysis. The results showed that 475 biological processes were significantly enriched in Wagyu cattle compared to Chinese Red Steppe cattle (Figure 1B). Among these enriched biological processes, seven (β-oxidation of fatty acids using acyl-CoA oxidase, regulation of fatty acid transport, metabolic processes of short-chain fatty acids, metabolic processes of fatty acids, biosynthetic processes of fatty acids, transport of plasma membrane long-chain fatty acids and negative regulation of fatty acid oxidation) were closely related to beef quality traits.

The results of KEGG enrichment analysis were shown in Figure 1C. The 138 DEGs were enriched in 171 KEGG pathways, of which 11 pathways were significantly en-riched (*p* < 0.05), such as Alanine, aspartate and glutamate metabolism (PATH:00250), ECM-receptor interaction (PATH:04512) and D-Arginine and D-Ornithine Metabolism (PATH:00472). Genes specifically expressed in Chinese Red Steppe cattle were mainly enriched in 20 pathways, which also included D-arginine and D-ornithine metabolism. In addition, genes specifically expressed in Wagyu were enriched in seven pathways (*p* < 0.05), with the highest enrichment in dilated cardiomyopathy and ECM–receptor interaction.

### 2.3. Prediction Analysis of Genes Interaction Network

The results of the gene interaction network prediction analysis are shown in Figure 2A. Among these, the down-regulated genes enoyl-CoA hydratase and 3-hydroxyacyl CoA dehydrogenase (*EHHADH*), the up-regulated gene *ACOX1* and the down-regulated genes aldehyde dehydrogenase 9 family member A1 (*ALDH9A1*) and acyl-CoA synthetase (*ACSS*) formed regulatory relationships through intermediate compounds, including trans-hexadec-2-enoyl-CoA, (S)-methylmalonate semialdehyde and acetoacetyl-CoA in: fatty acid degradation (bta00071); valine, leucine and isoleucine degradation (bta00280); and butanoate metabolism (bta00650), respectively. The up-regulated collagen type IV alpha 1 chain (*COL4A1*) gene may play a role in activating the up-regulated genes *SDC3* and *CD44* via ECM–receptor interaction (bta04512), and *SDC3* is also activated by the up-regulated genes thrombospondin 4 (*THBS4*); the up-regulated gene adenylate cyclase 5 (*ADCY5*) may interact with the up-regulated gene protein kinase cAMP-activated catalytic subunit beta (*PRKACB*) via several pathways, and it also has an interaction with the down-regulated gene phosphodiesterase 4C (*PDE4C*) via cyclic GMP in pathways of purine metabolism (bta00230). *PRKACB* acts as an activator of the up-regulated calcium voltage-gated channel auxiliary subunit gamma 4 (*CACNG4*) gene in dilated cardiomyopathy (bta05414); the down-regulated gene glutamic-oxaloacetic transaminase 1 (*GOT1*) is associated with the down-regulated genes adenylosuccinate synthase (*ADSS*) and D-aspartate oxidase (*DDO*). The down-regulated gene methionine adenosyltransferase 2B (*MAT2B*) and the up-regulated gene DNA methyltransferase 3 alpha (*DNMT3A*) form an interaction through S-adenosyl-L-methionine in cysteine and methionine metabolism (bta00270); the down-regulated genes thymidylate synthetase (*TYMS*) and the down-regulated gene methylenetetrahydrofolate dehydrogenase, cyclohydrolase and formyltetrahydrofolate synthetase 1 (*MTHFD1*) may interact with 5,10-methylene-tetrahydrofolate in one carbon pool via folate (bta00670).

### 2.4. Validation of Candidate DEGs Expression Levels

The expression levels of DEGs connected to lipid metabolism were verified using RT-qPCR. The mRNA levels of *ASIP*, GINS complex subunit 2 (*GINS2*), *FBP1*, phospholipase C epsilon 1 (*PLCE1*) and geminin DNA replication inhibitor (*GMNN*) genes in Chinese Red Steppe cattle were significantly lower than in Wagyu (*p* < 0.05), and the mRNA expression levels of scavenger receptor class F member 2 (*SCARF2*), *SDC3*, *CRYAB*, *ANKRD2*, dehydrogenase/reductase 3 (*DHRS3*), annexin A2 (*ANXA2*), murine retrovirus integration site 1 homolog (*MRVI1*) and NUMB like endocytic adaptor protein (*NUMBL*) were significantly higher in Wagyu (*p* < 0.05). The results of the RT-qPCR were consistent with the sequencing data (Figure 2B).

### 2.5. Effects of Candidate Genes on Triglyceride Content in Bovine Fetal Fibroblasts (BFFs)

To analyze the functions of key candidate DEGs involved in lipid metabolism, overexpression vectors of related genes were constructed separately. The coding sequences (CDSs) of the *ANKRD2*, *CRYAB*, *ALDH9A1* and *EHHADH* genes were successfully amplified by PCR and ligated to the pBI-CMV3 vector (Figure 3A). The BFFs in overexpression and negative control groups showed regular morphological expression of green fluorescent proteins (GFPs) 24 h post-transfection (Figure 3B). RT-qPCR was used to detect the candidate genes. The results showed that the mRNA expression levels of *ANKRD2*, *CRYAB*, *ALDH9A1* and *EHHADH* in each overexpression group significantly increased (*p* < 0.01) compared with the control group (Figure 3C).

Detection of intracellular triglyceride showed that the overexpression of *ANKRD2*, *ALDH9A1* and *EHHADH* resulted in significant decreases in the intracellular triglycerides (*p* < 0.01). We found the most pronounced decrease after the overexpression of *EHHADH* (33.132 ± 4.127 μmol/g). Overexpression of *CRYAB* in BFFs also lowered intracellular triglycerides (49.326 ± 0.341 μmol/g), whereas there was no significant difference between the pBI-CMV3-CRYAB and pBI-CMV3 groups (54.387 ± 0.791 μmol/g, Figure 3D). These results suggest that *CRYAB*, *ANKRD2*, *ALDH9A1* and *EHHADH* are negative regulators of triglyceride synthesis in BFFs. 

### 2.6. Effects of Candidate Genes on Bovine Intracellular Fatty Acids

The results of the gas chromatography analysis showed that six fatty acid components were detected in the cells: caproic, caprylic, palmitic, stearic, linoleic and cis-4,7,10,13,16,19-docosahexaenoic acids (Figure 4). Short-chain fatty acids (hexanoic) are critical nutrients for ruminants, and medium-chain fatty acids (octanoic) are required for regulation of neural energy balance. Palmitic acid is thought to promote the accumulation of triglycerides in livestock, stearic acid reduces low-density lipoprotein (LDL) cholesterol and linoleic acid is an essential fatty acid. Therefore, these fatty acids are of considerable interest for their nutritional and therapeutic properties.

Compared with the control group, each fatty acid (except for hexanoic acid) and total fatty acids decreased in the *CRYAB* and *ANKRD2* overexpression groups; the content of each fatty acid and total fatty acids within the *ALDH9A1* overexpression group (83.48 ± 20.22 μg) was higher than that for the control (34.29 ± 20.10 μg). In the *EHHADH* group, the amount of linoleic acid (1.79 ± 0.863 μg) and cis-4,7,10,13,16,19-docosahexaenoic acid (2.01 ± 0.686 μg) increased in BFFs, while the rest of the fatty acids decreased. In addition, hexanoic acid was not detected in the *ALDH9A1* and *EHHADH* overexpression groups.

### 2.7. Regulation of Genes Related to Lipid and Fatty Acid Metabolism by Candidate Genes

*ANKRD2*, *CRYAB*, *ALDH9A1* and *EHHADH* gene overexpression in BFFs was analyzed using the RT2 Profiler PCR Array to detect their effects on key genes involved in fat and fatty acid metabolism (Figure 5). Overexpression of *CRYAB* resulted in up-regulation of glycerol-3-phosphate dehydrogenase 1 (FC > 1.5, *p* < 0.05) and down-regulation of acyl-CoA synthetase long chain family member 6 (*ACSL6*), methylmalonyl-CoA epimerase (*MCEE*), protein kinase AMP-activated non-catalytic subunit gamma 1 (*PRKAG1*), lipoprotein lipase (*LPL*) and acetyl-CoA acetyltransferase 1 (FC < 1/1.5, *p* < 0.05). 

As a result of *ANKRD2* overexpression, the levels of 3-hydroxy-3-methylglutaryl-CoA synthase 2 (*HMGCS2*) and methylmalonyl CoA mutase (*MUT*) were enhanced (FC > 1.5, *p* < 0.05), whereas glycerol kinase 2 (*GK2*) expression levels were reduced (FC < 1/1.5, *p* < 0.05). 

Upon overexpression of *ALDH9A1*, fatty acid binding protein 5 (*FABP5*), solute carrier family 27 member 1 (*SLC27A1*), acyl-CoA dehydrogenase short/branched chain (*ACADSB*), acyl-CoA dehydrogenase family member 11 (*ACAD11*), fatty acid binding protein 4 (*FABP4*), acyl-CoA synthetase long chain family member 5 (*ACSL5*), carnitine O-acetyltransferase (*CRAT*), peroxisomal trans-2-enoyl-CoA reductase (*PECR*), *HMGCS2* and *MUT* expression levels went up (FC > 1.5, *p* < 0.05), and lipase E, hormone-sensitive type (HSL/LIPE) was down-regulated (FC < 1/1.5, *p* < 0.05). 

*EHHADH* gene overexpression resulted in enoyl-CoA delta isomerase 2 (*ECI2*), glutaryl-CoA dehydrogenase (*GCDH*), inorganic pyrophosphatase 1 (*PPA1*), *FABP4*, acyl-CoA thioesterase 7 (*ACOT7*), 3-hydroxy-3-methylglutaryl-CoA lyase (*HMGCL*) and *HMGCS2* expression levels being up-regulated (FC > 1.5, *p* < 0.05) and 28 genes being down-regulated (FC < 1/1.5, *p* < 0.05).

Genes that are important for fat and fatty acid metabolism can be divided into four types. *ACAT1*, *GK2*, *SLC27A1* and *PPA1* promote the synthesis of triglycerides. Conversely, glycerol-3-phosphate dehydrogenase 1 (*GPD1*), *LPL* and *HMGCS2* promote triglyceride catabolism. In addition, *ACSL5*, *ACSL6*, *PRKAG1*, *FABP4*, *FABP5* and *ACOT7* promote fatty acid synthesis, whereas *MCEE*, *ACADSB*, *ACAD11*, *PECR*, *HMGCL* and *ECI2* promote fatty acid catabolism.

The protein interaction network between the candidate genes and the regulated fatty acid metabolism pathway genes was predicted and analyzed using STRING. The results showed that *ANKRD2* did not have protein interactions with up- and down-regulated genes. The *ALDH9A1* gene interacted with *MUT* (Figure 6B). The *EHHADH* gene had direct and indirect interactions with 28 genes; among the down-regulated genes, *HSL/LIPE* interacted with protein kinase AMP-activated non-catalytic subunit beta 2 (*PRKAB2*) and *PRKACB*. *GPD1* interacted with glycerol-3-phosphate dehydrogenase 2 (*GPD2*) (Figure 6C).

## 3. Discussion

Indicators such as fat deposition and fatty acid content play crucial roles in related economic traits, such as growth, reproduction and meat quality in livestock. The intramuscular fat content of beef affects tenderness, flavor, marbling and nutritional value. A moderate amount of intramuscular fat deposition can increase the marbling level, reduce the shear force of beef and improve the flavor [19]. Beef is rich in fatty acids and essential polyunsaturated fatty acids, and the type of fatty acid and unsaturated fatty-acids have also become important indicators for evaluating beef quality [20]. 

In the present study, the DEGs involved in growth and development and lipid metabolism were screened as candidate genes for bovine meat quality traits, including *CD44*, *SDC3*, *ASIP*, *CRYAB*, *ANKRD2*, *GOT1*, *SLC27A5*, *ALDH9A1*, *EHHADH*, *PECR*, *AGRN*, *ADCY5*, *ACOX1*, *COL4A1* and *ACTC1*. 

Of the DEGs, the *ASIP* gene has been reported to regulate coat color phenotype in animals [21]. Furthermore, with the discovery of the expression of *ASIP* mRNA in subcutaneous fat cells [22], the role of *ASIP* genes in lipid metabolism has become a subject of interest. Elke et al. [23] reported that the *ASIP* gene is widely expressed not only in adipose tissue but also in the muscle tissue of cattle. Compared with DEGs in Holstein cattle, *ASIP* mRNA was up-regulated more than nine-fold in the intramuscular fat of Japanese Wagyu cattle (*p* < 0.001), which suggested that it may be a functional gene related to the excellent beef quality of this breed. 

This study showed that the expression level of *AGRN* in the *Longissimus dorsi* of Wagyu was 1.79 times higher than that of Chinese Red Steppe cattle. Epigenetic studies also suggest a role for *AGRN* in human obesity as a link between DNA methylation and *AGRN* was found in a study involving obese and healthy individuals [24]. Meanwhile, our previous study found that the expression levels of *AGRN* were negatively correlated with its promoter DNA methylation levels [11]. Furthermore, Maak et al. [25] found that 11 genes were expressed in *Longissimus dorsi* samples with higher fat content in the F2 generation of a Charolais × Holstein cross. The up-regulated genes included *AGRN*, which indicates that it may be one of the candidate genes that increases fat deposition. 

Hematopoietic and immune cells highly express the *CD44* protein as a hyaluronan-binding surface receptor [26,27]. It is also highly expressed on the surface of human adipose stem cells, suggesting that *CD44* may be involved in the pluripotency and differentiation of preadipocytes [28]. Previous studies have shown that treating high-fat-diet mice with a monoclonal antibody of *CD44* suppresses the development of obesity and reduces adipose tissue inflammation [29]. Our recent study showed that the *CD44* gene is a key regulator of lipid metabolism in bovine mammary epithelial cells [30]. The above evidence suggests that *CD44* is important for lipid metabolism. 

The *SDC3* gene is thought to be a novel regulator of feeding behavior and body weight that participates in energy metabolism. It has been previously demonstrated that male *SDC3*-deficient mice respond to food deprivation by reflexively reducing their feeding [31]. In the presence of high-fat diets, *SDC3*-null mice accumulated less fat, demonstrated better glucose tolerance, and were resistant to obesity induced by high-fat diets [32]. In humans, *SDC3* polymorphisms have been linked to obesity and female hyperandrogenemia [33]. Interestingly, recent studies have found that SNPs in the *SDC3* gene are associated with growth traits in cattle [34]. This evidence suggests that *SDC3* may affect the formation of meat traits by regulating energy metabolism in domestic animals.

The expression level of *CRYAB* in the *Longissimus dorsi* of Wagyu was 2.57 times higher than that of the Chinese Red Steppe cattle (*p* < 0.05) in the present study. The expression level of the *CRYAB* gene in 3-month-old Japanese Wagyu × Hereford cattle was 1.73 times higher in Pyrmont × Hereford cattle, and the intermuscular fat content of Japanese Wagyu × Hereford hybrid cattle was significantly higher than that of the Pyrmont × Hereford hybrid cattle [35], suggesting that *CRYAB* may be related to multiple traits, such as high intermuscular fat content and muscle development. However, the overexpression of *CRYAB* inhibited fat deposition and reduced intracellular fatty acid content, which is inconsistent with the results in which the expression level of *CRYAB* was proportional to fat content. It is suggested that the effect of *CRYAB* in vivo on intermuscular fat deposition and lipid metabolism may occur through the coregulation of endocrine and paracrine pathways. Thus, gene overexpression could not increase the intracellular fat content in fibroblasts and might not be entirely consistent with the result in vivo. Hence, the regulatory effects of this gene on lipid metabolism and networks need further in vivo validation. In addition, *CRYAB* overexpression up-regulated *GPD1* (FC > 1.5, *p* < 0.05), and in a human obesity study, triglyceride synthesis was reduced and muscle mass was increased by growth factor receptor bound protein 14 (*GRB14*) and *GPD1*. The expression levels of *GPD1* and *GDF8* were down-regulated after weight loss, but increased in obese women compared with lean women [36]. At the same time, studies have also shown that in the omental adipose tissue of obese women, *LPL*, *GPD1* and leptin (*LEP*) are significantly reduced, suggesting that the increased expression of *GPD1* promotes fat deposition [37]. The down-regulated gene *PRKAG1* was located in a quantitative trait locus (QTL) associated with pig fat traits, suggesting that the *PRKAG1* gene may also be associated with bovine fat traits. The above results suggest that *CRYAB*-mediated regulation of fat content has an effect on fat deposition capacity, but its effects on fat deposition and the regulatory mechanisms involved need further analysis in vivo.

*ANKRD2* is localized to the nucleus and sarcomere in muscle cells and plays a role in the differentiation of muscle cells, as shown by its expression being induced during *C2C12* differentiation in vitro [38]. The results of this study showed that *ANKRD2* gene overexpression inhibited intracellular fat deposition, reduced intracellular fatty acid composition and regulated expression levels of genes in fatty acid metabolism pathways. At the same time, studies in a diabetic mouse model showed that the expression level of *ANKRD2* was also changed in diabetes [39]. Overexpression of *ANKRD2* could up-regulate *HMGCS2* and down-regulate *GK2*. Previous studies found that *HMGCS2* induced fatty acid α-oxidation and ketone production in hepatoma cells and played a crucial role in fatty acid oxidation [40]. In addition, *HMGCS2* up-regulation increased intracellular fat oxidation and reduced triglyceride content [40]. It is worth noting that *GK2* was found to be associated with pig backfat thickness in a genome-wide association analysis of Duroc pigs [41]. Overall, the *ANKRD2* gene could improve meat quality traits by regulating the expression of genes such as *HMGCS2* and *GK2*, which are involved in glucose and lipid metabolism, to affect intracellular fat and fatty acid composition.

Transcriptome analysis showed that the expression level of the *ALDH9A1* gene in the *Longissimus dorsi* of Wagyu was 62.93% of the level in Chinese Red Steppe cattle. We speculate that the *ALDH9A1* gene is also a potential functional gene for meat quality traits in cattle. In a study in which blood lipids were reduced in rats fed mulberry leaves, the expression level of *ALDH9A1* was significantly lower in these rats, suggesting that *ALDH9A1* also participates in lipid metabolism [42]. Notably, genome-wide association analysis in multiple pig populations found that *ALDH9A1* was associated with fatty-acids in muscle and abdominal adipose tissue in pigs [43], suggesting a potential role for *ALDH9A1* in meat quality traits and fatty acid content in pork [44]. Furthermore, overexpression of *ALDH9A1* could result in up-regulation of the *ACADSB* expression level (FC > 1.5, *p* < 0.05). Our previous studies found that the *ACADSB* gene could significantly increase the intracellular triglyceride content (*p* < 0.05) [45] and that the knockout of *ACADSB* in bovine mammary epithelial cells may have been an important regulator of intracellular fatty acid content [46]. However, we found that overexpression of the *ALDH9A1* gene caused a significant decrease in intracellular triglyceride content (*p* < 0.05), suggesting that it may not regulate intracellular triglyceride contents through increased *ACADSB* gene expression alone.

We found that the expression level of *EHHADH* in the *Longissimus dorsi* muscle of Wagyu was only 25.33% of the expression level of Chinese Red Steppe cattle (*p* < 0.05). *EHHADH* is part of the canonical peroxisomal fatty acid β-oxidation pathway that can be induced by *PPARα* activation [47]. In a dairy cow genome-wide association analysis study, *EHHADH* and 19 other genes were correlated with milk fatty acid traits in a Chinese Holstein dairy cow population [48], suggesting that *EHHADH* is a potential functional gene that affects fatty acid metabolism in cattle. Overexpression of *EHHADH* in BFFs significantly reduced intracellular triglycerides. Meanwhile, it resulted in up-regulation of seven genes and down-regulation of 28 genes in the fatty acid metabolism pathway, among which the expression level of carnitine palmitoyltransferase 1A (*CPT1A*) was down-regulated by a factor of 57.28. *CPT1* is a key rate-limiting enzyme responsible for the transport of long-chain fatty acids into mitochondria, and overexpression of *CPT1* in skeletal muscle in vivo increases fatty acid oxidation and reduces triacylglycerol esterification [49]. Stimulation of systemic *CPT1* activity may also accelerate peripheral fatty acid oxidation [50]. Furthermore, *CPT1* has three paralogous genes in mammals: *CPT1A*, *B*, and *C*. *CPT1A* is mainly expressed in the liver, whereas *CPT1B* is expressed in muscle and, to a lesser degree, in the liver. After treatment with tetradecylglycine, *PPARα*, *CPT1A* and *CPT1B* were significantly up-regulated in the livers of mice [51]. It appears that the expression of *CPT1A* and *CPT1B* may vary with the timing of *PPARα* activation or may not be fully mediated by the *PPARα* pathway. Notably, it was found in our previous study that *HSL* gene overexpression led to up-regulation of the *EHHADH* gene [52], whereas overexpression of the *EHHADH* gene resulted in down-regulation of *HSL* gene expression. Although both *HSL* and *EHHADH* are essential genes in regulation of fat and fatty acid metabolism, the regulatory roles of *HSL* and *EHHADH* are poorly defined [53,54,55]. From the results of our previous study, it is speculated that *HSL* overexpression increased intracellular triglyceride content, so the oxidation of *EHHADH* was activated. However, *EHHADH* gene overexpression led to a decrease in intracellular triglyceride in this study. We believe that with decreasing triglyceride levels, *HSL* gene expression became inactive and the decreasing content of triglycerides reduced its expression. Therefore, the regulatory mechanism of the level of expression and the interaction between *HSL* and *EHHADH* need further study.

## 4. Materials and Methods

### 4.1. Animals and Longissimus dorsi Sample

The *Longissimus dorsi* of Wagyu and Chinese Red Steppe used in this experiment were provided by Inner Mongolia University (Hohhot, China) and Agricultural Science Academy of Jilin Province (Gongzhuling, China), respectively. The two farms raising the two groups of cattle are located at similar altitudes and have similar natural climatic conditions. Cattle in both groups were grown under similar feeding conditions and fed on corn and hay with free access to fodder and water. The Wagyu and three Chinese Red Steppe were slaughtered at 28 months for longissimus dorsi muscle tissues, and biological replicates are three. All the tissue details were shown in our previous study [11]. Cut the samples into pieces, aliquot them within cryovials, and store them in liquid nitrogen quickly. All the animal experiments in the present study strictly complied with the relevant regulations regarding the care and use of experimental animals issued by Jilin University Animal Care and Use Committee (Approval ID: 20140310).

### 4.2. RNA Extraction and RNA Sequencing

Total RNAs of tissues were extracted by Trizol (Takara, Dalian, China). Extractions were treated with DNaseI (NEB, Beijing, China), and the concentration of total RNA obtained was detected by Agilent 2100 Bioanalyzer (Davis, CA, USA). The purified RNA was then used for RNA sequencing. First, enrichment of eukaryotic mRNA with magnetic beads with Oligo (dT). Then, the mRNA is broken into short fragments by adding a breaking reagent at the appropriate temperature in the thermomixer (Eppendorf AG, Hamburg, Germany). The broken mRNA is used as a template to synthesize cDNA and then synthesize the second cDNA. Finally, the product is purified and recovered, the sticky ends are repaired and base “A” is added to the 3′ end of cDNA to connect the linker. The constructed library was sequenced using the Illumina HiSeq2000 (BGI, Shenzhen, China) sequencing platform after quality checking using ABI Step One Plus Real-Time PCR System (StepOnePlus; Applied Biosystems, Waltham, MA, USA).

### 4.3. Sequencing Data Analysis

The original sequencing data is called raw reads, and clean reads are obtained after filtering out low-quality reads. The clean reads were mapped to the reference genome (*Bos taurus* UMD_3.1.1) using HISAT2 software [56]. Next, we used Bowtie2 [57] to align clean reads to this reference sequence and then used RSEM [58] to calculate gene and transcript expression levels. We use the internationally recognized algorithm EBSeq to differentially base Fragments Screening of factors (Log2^FC^ > 0.585 or <−0.585, FDR < 0.05). 

### 4.4. Gene Ontology Enrichment and Kyoto Encyclopedia of Genes and Genomes Pathway Analysis

The DEGs were subjected to GO enrichment analysis by the R language package of GO seq [59], with gene length corrected for bias. KOBAS software [60] was used to test the statistical enrichment of related genes in the KEGG pathways [61]. A corrected *p* < 0.05 was considered significant enrichment.

### 4.5. Real-Time Quantitative PCR Analysis

To detect their relative expression levels of mRNA by Real-Time Quantitative PCR (RT-qPCR), the primers for RT-qPCR were designed using Primer Premier 6.0 software. The details of the gene ID and primer sequences are shown in Appendix A. The β-actin (*ACTB*) gene was selected as the internal reference gene. RT-qPCR reaction system: SYBR^®^ Premix Ex Taq (Tli RNaseH Plus) (2×) 5.0 μL, PCR upstream and downstream primers (10 μmol/L) each 0.2 μL, cDNA template 1.0 μL, nuclease-free water 3.6 μL, total reaction volume 10.0 μL; reaction conditions: 95 °C for 5 min; 95 °C for 10 s, 60 °C for 30 s, 40 cycles; add a melting curve program after quantifying the amplification parameters: 95 °C—15 s, 60 °C—20 s, 95 °C—15 s. 

### 4.6. Construction of Candidate Gene Overexpression Vector

Two pairs of primers were designed for each candidate gene, and the CDS region sequence of the target fragment was amplified by nested PCR. The details of the primer sequences and mRNA ID are shown in Appendix A. PCR products were verified by Sanger sequencing and ligated into pBI-CMV3 plasmid (#631632, Clontech Laboratories, Mountain View, CA, USA) to generate overexpression vectors for candidate genes.

### 4.7. Cell Culture and Transfection

The BFFs in the study were purified and cultured from newborn cattle ear tip tissues according to our laboratory’s previous method using tissue nubble culture [62]. The BFFs were cultured in 10 cm culture plates (Falcon, 353003, Franklin, Lake, NJ, USA) in DMEM/F12 (HyClone, 12-719Q, Logan, UT, USA) supplemented with 10% FBS (Fetal bovine serum, 11011-6123, Tian Hang, Hangzhou, Zhejiang, China). To investigate the role and regulatory mechanism of candidate genes for meat quality traits on fat and fatty acid metabolism, we seeded cells at a concentration of 2 × 10^6^ cells/well in six-well culture plates (353,090, Falcon) and cultured at 37 °C and 5% CO_2_ in an incubator (Thermo Fisher Scientific, Inc., Waltham, MA, USA). When the density of cells reached 80%, exchange the culture medium for transfection. Each overexpression vector of candidate genes was transfected into cells using FuGENE HD Transfection Reagent (PRE2311, Promega, Madison, WI, USA) according to the manufacturer’s protocol, respectively, and the group of BFFs transfected with pBI-CMV3 was the negative control group. At 24 h post-transfection, cell morphology and growth state were observed under microscope, and the expression of green fluorescent protein (GFP) in the cells was observed under a fluorescence microscope (Nikon TE2000, Tokyo, Japan) to observe the transfection efficiency. Triplicate experiments were performed by transfecting the same number of cells with the same vector in different wells.

### 4.8. Analysis of the Triglyceride Contents in BFFs

Cells were collected post-transfection for 48 h. The triglyceride contents in each group of BFFs were detected according to the manufacturer’s protocol of triglyceride assay kit (Applygen Technologies, E1015-105, Beijing, China), and the absorbance value at the wavelength of 550 nm was detected by a multi-function microplate reader (Biotech, San Francisco, CA, USA). The BCA protein detection kit (KeyGEN BioTECH, KGP902, Nanjing, Jiangsu, China) was used, and the test steps were referred to the standard operating procedure of the instruction manual to detect the concentration of protein in the same sample. The triglyceride content was finally corrected per mg protein content.

### 4.9. Fatty Acids (FAs) Extraction and Content Analysis in BFFs by GC

FAs were extracted post-transfection for 48 h of BFFs. After washing 3 times with phosphate-buffered saline (PBS), cells were trypsinized with 0.25% trypsin solution (Gibico, Grand Island, NY, USA). Cell pellets were collected by centrifugation at 500 g. The methods of FAs detection were referred to Pingjiang [30]. Briefly, Folch solution (2:1 CHCl3:CH3OH, *v*/*v*) and internal reference FA (Ginkgolic acid C13:0, 49,962, Sigma-Aldrich, St. Louis, MO, USA) were added to cells of each group. Then, the tube with the cells pellet was filled with high-purity nitrogen and shaken vigorously. After chloroform was evaporated, methylated mixed solvent consisting of 35% BF3 methanol (14%) (33040-U, Sigma-Aldrich), 45% methanol and 20% hexane was added to the glass tube. Finally, 1 mL of hexane and 0.4 mL of NaCl (0.88%) were added, and the supernatant was transferred via a Pasteur pipette with a long pipette into a clean glass vial with a lid for gas chromatography analysis (GC7980, ALS7020, Techcomp, Hong Kong, China). The standard FAs (Supelco37, 18919-1 AMP, Sigma-Aldrich) is the standard solution. The Gas Chromatograph System (Agilent 7890A) was used with an HP-FFAP elastic quartz capillary column (100 m × 0.25 mm, film thickness 0.2-μm) (CP-Sil 88 for Methyl esters, Agilent, Santa Clara, CA, USA) to detect fatty acid methyl ester. The specific operating conditions are as follows. The initial column temperature is set to 70 °C. The injection and detector temperatures were 250 °C and 255 °C, respectively. The split ratio was 10:1 and the carrier gas was nitrogen. The injection volume under operating conditions was 1.0 μL. The flow rates of hydrogen, nitrogen, and air gases at the outlet were 25, 20 and 150 mL/min, respectively. Clean after every four samples. The content of FA components was calculated by peak area normalization. Thirty-seven standard FAs are measured and the content of the FA should be calculated according to the amount of the internal reference FA and the ratio of FA to the total FA. 

### 4.10. Analysis of Lipid Metabolism by RT^2^ Profiler PCR Array

Two micrograms of total RNA were extracted from BFFs using the RNeasy mini kit (74134, Qiagen, Frederick, MD, USA). The cDNA was then synthesized using the RT2 First Strand Kit (330404, Qiagen, Frederick, MD, USA) according to the manufacturer’s protocol. RT-qPCR was performed with an Mx3005p system (Stratagene, Agilent, Santa Clara, CA, USA). The transcript levels of lipid metabolism genes were detected by RT2 profile PCR Array (CLAB24070A, Qiagen, Frederick, MD, USA). According to the manufacturer’s instructions, RT2 SYBR Green ROX qPCR Master Mix (Qiagen, Frederick, MD, USA) was used. The reactions were incubated at 95 °C for 10 min, followed by 40 cycles of 95 °C for 10 s and 60 °C for 1 min, then add a melting curve program: 95 °C 15 s, 60 °C 20 s and 95 °C 15 s. *β-actin*, glyceraldehyde-3-phosphate dehydrogenase (*GAPDH*), tyrosine hydroxylase (*YWHAZ*), hypoxanthine phosphoribosyltransferase (*HPRT1*) and TATA-binding protein (*TBP*) is the reference control. Reference genes were selected based on normalized threshold count (CT) values. The online RT2 Profiler PCR Array data analysis software (https://geneglobe.qiagen.com/cn/analyze, accessed on 1 May 2021) was used to analyze the relative gene expression data). *p* value < 0.05 was considered statistically significant.

### 4.11. Statistical Analysis

Experimental results are expressed as mean ± standard error of measurement (SEM). Relative expression levels of DEGs were calculated using the comparative Ct method (2^−ΔCt^). Meanwhile, candidate genes expression levels of overexpression groups and the negative control group were calculated using the comparative Ct method (2^−ΔΔCt^). The expression level of each mRNA relative to *β-actin* was analyzed and calculated. GraphPad Prism 6 software (GraphPad Software, San Diego, CA, USA) was used to analyze the data with a *t*-test. The statistical analysis of triglyceride contents was performed with GraphPad Prism 6 software and carried out by one-way analysis of variance (ANOVA) with Dunnett’s multiple comparisons tests. *p* < 0.05 was defined as statistical significance.

## 5. Conclusions

This study generated a dataset from transcriptome profiling of two cattle breeds with differing fat deposition capacities of the muscle and identified 388 DEGs in the *Longissimus dorsi* between Wagyu and Chinese Red Steppe. The presented DEGs confirm in part previous reports about the functional genes related to intramuscular fat deposition and also provide novel candidate genes related to meat quality traits in cattle. Meanwhile, *CRYAB*, *ANKRD2*, *ALDH9A1* and *EHHADH* gene overexpression inhibited intracellular triglycerides and affected intracellular fatty acid components by regulating gene expression levels in fat and fatty acid metabolic pathways. The results provide valuable insight into the significant variation between Wagyu and Chinese Red Steppe cattle meat quality and offer useful genetic markers for the breeding of high-grade beef. 

## Figures and Tables

**Figure 1 ijms-24-00387-f001:**
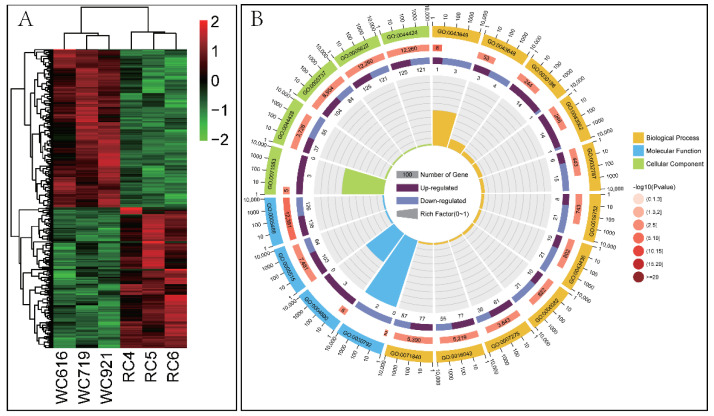
Differential mRNAs expression in *Longissimus dorsi* muscles between Japanese black cattle (Wagyu) and Chinese Red Steppe cattle. (**A**) Heat map of differentially expressed genes (DEGs); (**B**) the top 20 Gene Ontology (GO) terms by ascending order of corrected *p* value (BP: biological process; MF: molecular function; CC: cell composition); (**C**) the top 20 KEGG pathways listed by ascending order of corrected *p*-value.

**Figure 2 ijms-24-00387-f002:**
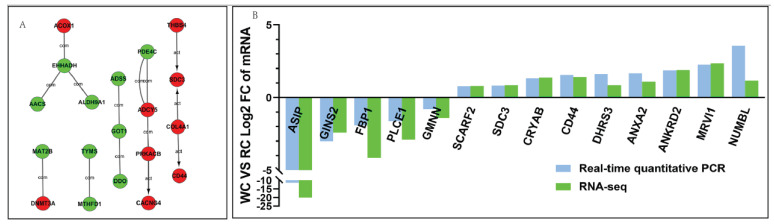
Prediction analysis of genes interaction network and validation of sequencing data. (**A**) Gene interaction network: red circles represent up-regulated genes, green circles represent down-regulated genes; (**B**) the fold change of DEG expression levels.

**Figure 3 ijms-24-00387-f003:**
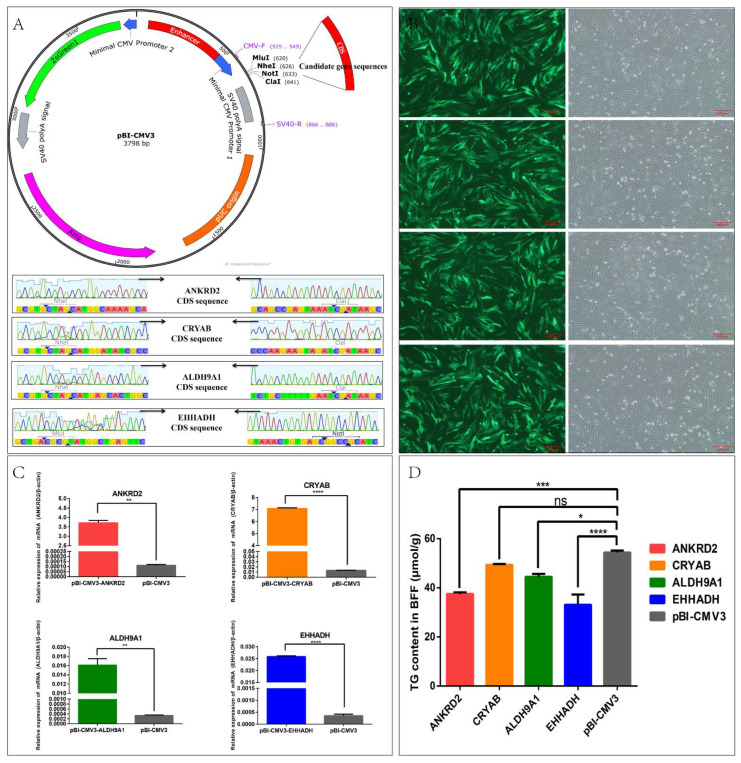
Construction of candidate gene overexpression vectors and their effects on triglyceride content in bovine fetal fibroblasts (BFFs). (**A**) The coding sequence (CDS) of the candidate gene was amplified by PCR and ligated to the pBI-CMV3 vector; (**B**) green fluorescent protein expression of BFFs 48 h after transfection; (**C**) RT-qPCR to detect mRNA levels of candidate genes; (**D**) detection of triglyceride content in BFFs. * *p* < 0.05, ** *p* < 0.01, *** *p* < 0.001, **** *p* < 0.0001.

**Figure 4 ijms-24-00387-f004:**
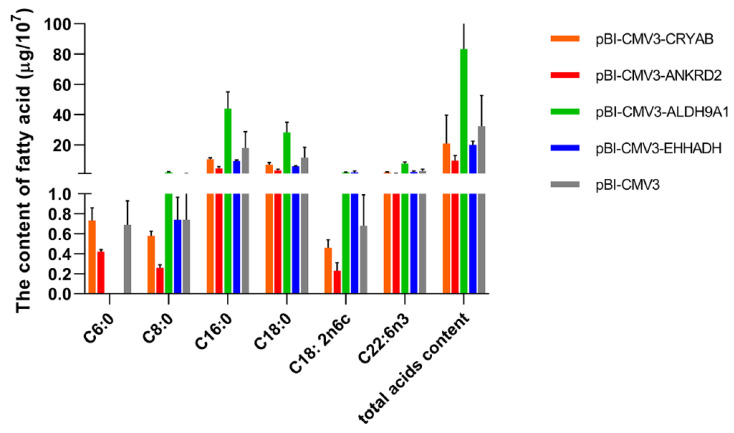
The composition of fatty acids 48 h after transfection. C6:0: caproic acid; C8:0: caprylic acid; C16:0: palmitic acid; C18:0: stearic acid; C18:2n6c: linoleic acid; C22:6n3: cis-4,7,10,13,16,19-docosahexaenoic acid.

**Figure 5 ijms-24-00387-f005:**
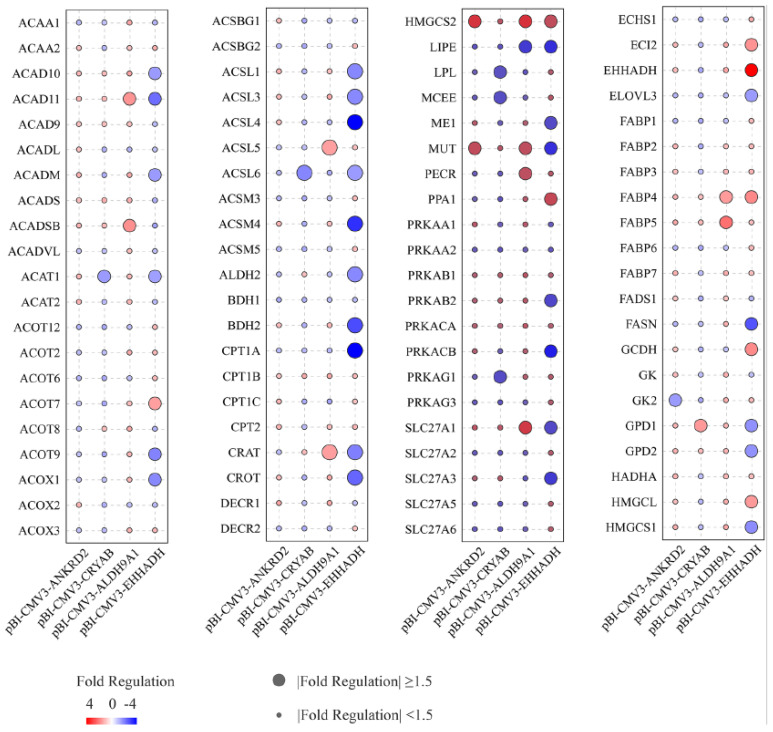
Regulation of genes related to fat and fatty acid metabolism by candidate genes.

**Figure 6 ijms-24-00387-f006:**
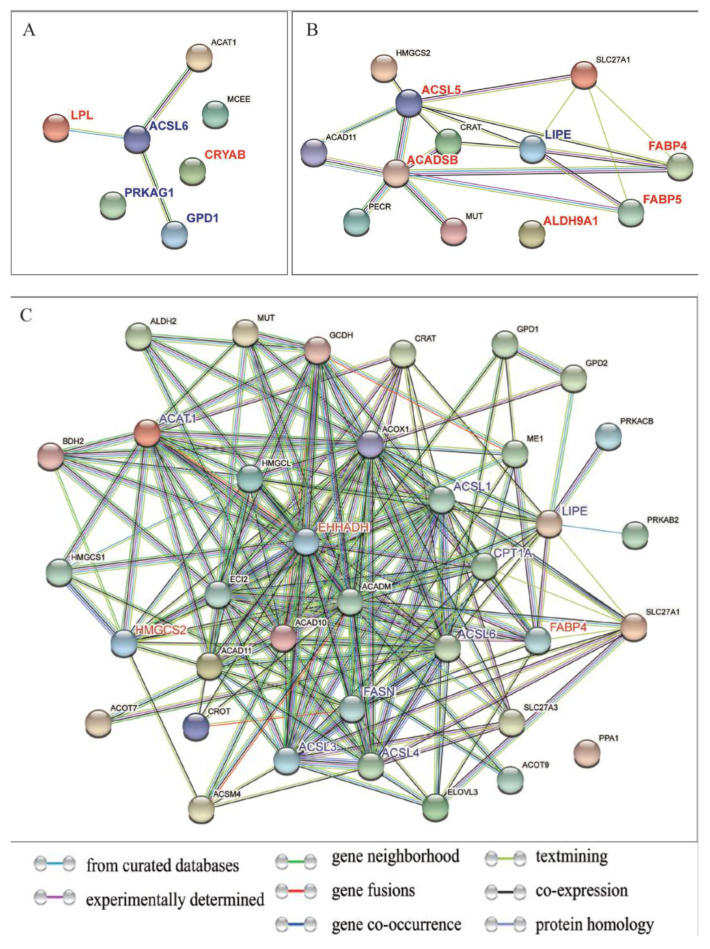
Prediction of protein interaction between genes related to fat and fatty acid metabolism and candidate genes. (**A**) CRYAB and fatty acid metabolic pathway protein interaction network; (**B**) ALDH9A1 and fatty acid metabolic pathway protein interaction network; (**C**) EHHADH and fatty acid metabolic pathway protein interaction network.

**Table 1 ijms-24-00387-t001:** The top 10 up-regulated DEGs in *Longissimus dorsi* in Wagyu and Chinese Red Steppe cattle.

Gene	EnsenbleID	Wagyu	Chinese Red Steppe	Log2FC	FDR
*CLDN7*	ENSBTAG00000019448	38.11	0.68	5.80	0.00
*ENSBTAG00000023318*	ENSBTAG00000023318	15.24	0.68	4.48	0.00
*ACTC1*	ENSBTAG00000005714	1727.76	92.25	4.23	0.04
*CCL22*	ENSBTAG00000017718	17.49	1.36	3.68	0.00
*HUNK*	ENSBTAG00000020762	29.78	2.39	3.64	0.00
*FBN3*	ENSBTAG00000000595	23.37	3.07	2.93	0.02
*KCNC3*	ENSBTAG00000039935	15.62	2.39	2.71	0.01
*RFX2*	ENSBTAG00000017661	195.96	30.82	2.67	0.00
*RIMBP2*	ENSBTAG00000009797	14.83	2.40	2.63	0.00
*CDS1*	ENSBTAG00000045787	10.62	1.72	2.62	0.03

DEGs: differentially expressed genes; FC: fold change; FDR: false discovery rate.

**Table 2 ijms-24-00387-t002:** The top 10 down-regulated DEGs in *Longissimus dorsi* in Wagyu and Chinese Red Steppe cattle.

Gene	EnsenbleID	Wagyu	Chinese Red Steppe	Log2FC	FDR
*ASIP*	ENSBTAG00000034077	0.00	165.14	−20.00	0.00
*CLRN2*	ENSBTAG00000003357	0.00	133.14	−20.00	0.00
*ENSBTAG00000021906*	ENSBTAG00000021906	0.00	40.38	−20.00	0.00
*PRM1*	ENSBTAG00000021493	0.00	25.86	−20.00	0.00
*HBB*	ENSBTAG00000038748	1.01	134.73	−7.06	0.00
*DNER*	ENSBTAG00000016063	2.17	93.40	−5.42	0.00
*IRX5*	ENSBTAG00000004838	1.95	82.41	−5.40	0.00
*HP*	ENSBTAG00000006354	0.31	8.25	−4.74	0.00
*FBP1*	ENSBTAG00000009733	6.89	121.92	−4.15	0.04
*MPZ*	ENSBTAG00000033835	15.76	214.42	−3.77	0.02

DEGs: differentially expressed genes; FC: fold change; FDR: false discovery rate.

## Data Availability

The datasets presented in this study can be found in online repositories. The names of the repository/repositories and accession number(s) can be found at: https://www.ncbi.nlm.nih.gov/genbank/ (accessed on 1 May 2022), GSE161967.

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
