# Peer review of "RNA-Seq Analysis Identifies Differentially Expressed Genes in the Longissimus dorsi of Wagyu and Chinese Red Steppe Cattle"

_ijms, 2022, doi:10.3390/ijms24010387_

Round 1

Reviewer 1 Report

1. Some sentences are not correctly framed. 

2. The discussion section is a bit too long and can be reduced. 

3. A comprehensive schematic would help.

Reviewer 2 Report

The manuscript entitled: RNA-Seq Analysis Identify Differentially Expressed Genes in the Longissimus Dorsi of Wagyu and Chinese Red Steppes cattle addresses focused on an interesting research topic in animal science and molecular animal breeding fields. However, the experimental design is not clear. The materials and methods section is not well described and a lot of valuable information is missing. The English language requires revision by a native speaker. The authors have used the standard format in manuscript writing. Overall, the manuscript is within the scope focused of the International Journal of Molecular Sciences. However, the manuscript requires major revision before taking a decision.

Reviewer 3 Report

Through RNA-seq analysis, the author found DEGs in the longissimus dorsi of  Wagyu and Chinese Red Steppes providing some new candidate genes related to beef quality traits. The results provide valuable reference for the study of the significant differences in beef quality between  Wagyu and Chinese Red Steppes. It provides a large number of potential genes and genetic basis for further exploring the functional genes affecting beef quality traits. However, the manuscript still needs extensive revision.

1. There are a lot of format and language problems in the manuscript. The annotation is not clear. Doesn't the author know that genes need italics?

2.The up and down DEG of table1 can be displayed separately.

3.Line114, the prediction network indicates more potential co expression relationships. There is no indicated correlation with the production compound. Please list the relevant literature.

4.In the method, it was transfected for 48 hours, but in the result, the fluorescent protein was observed 24 hours after transfection. Please confirm?

5.153 lines, please mark the p value represented by "* * *"

6.In figure 3, do overexpression cells detect TG after adipogenic differentiation? According to the research results, after gene overexpression, TG content is affected to varying degrees. Whether oil red O staining leading to fat differentiation can be added for additional explanation, because there is no fat induction, TG content determination will have an impact.

7.Line 163, what statistical method is used for fat determination?

8.Line 186, "figure 4", are you sure it is not "figure 5"?

9.Line 231, "The composition of intramuscular fat and fatty acid in beef affects the meat flavor". Since the author has studied the muscle DEG of two different kinds of cattle, and constructed multiple DEG overexpression vectors to study the effects on TG and fatty acid content in bovine muscle fiber cells, why not directly compare the muscle fiber types, intramuscular fat content, TG content and fatty acid composition of two different beef tissues, and screen DEG? And further explore the correlation between DEG and phenotypic data?

10.Lines 250 to 254 need to reorganize the language.

11. In the manuscript, the author discussed the functions of a large number of different genes. Please sum up and draw what conclusions?

12. 509 lines and 511 lines, please mark uniformly.

13. 530 lines delete spaces, and“poultry”.

Round 2

Reviewer 2 Report

The manuscript has been improved significantly to the level of recommending its acceptance. 

Author Response

Thank you very much for your review of our manuscript.

Reviewer 3 Report

有必要添加全文摘要,以引导读者全面理解文章。
